Large-sized fossil hamsters from the late Middle Pleistocene Locality 2 of Shanyangzhai, China, and discussion on the validity of Cricetinus and C. varians (Rodentia: Cricetidae)

Xie Kun
Zhang Yunxiang yxzhang@nwu.edu.cn
Li Yongxiang
State Key Laboratory of Continental Dynamics, Department of Geology, Northwest University , Xi’an, Shaanxi Province , China
De Baets Kenneth
Electronic publication date: 2023 Jul 28
Publication date: 2023
Volume: 11
Electronic Location ID: e15604
Received 2023 Jan 7; Accepted 2023 May 31
Copyright: © 2023 Xie et al.
Copyright year: 2023
Copyright holder: Xie et al.
License: This is an open access article distributed under the terms of the Creative Commons Attribution License, which permits unrestricted use, distribution, reproduction and adaptation in any medium and for any purpose provided that it is properly attributed. For attribution, the original author(s), title, publication source (PeerJ) and either DOI or URL of the article must be cited.
License URL: https://creativecommons.org/licenses/by/4.0/

Keywords: Shanyangzhai fauna, Cricetinus varians, Tscherskia triton varians, Neocricetodon

Funding: National Natural Science Foundation of China 42172010 This work was supported by the National Natural Science Foundation of China (No. 42172010). The funders had no role in study design, data collection and analysis, decision to publish, or preparation of the manuscript.

==============================
A detailed morphological description and comparative study were conducted on numerous large-sized hamster remains collected from the late Middle Pleistocene Locality 2 of Shanyangzhai (Syz 2), Hebei Province, China. The comparisons reveal that these fossils are highly similar to the extant Tscherskia triton in size and morphology, including the small degree of alternating between the main opposite cusps on M1-3, the presence of axioloph on M3, and mesolophids on m1-2 that are present but seldom reach the lingual margin of the teeth, among other features. However, minor differences between the two still exist. Consequently, all these fossils are designated as a chronosubspecies of the extant species, T. triton varians comb. nov. The skull and molar morphologies of Cricetinus varians and T. triton were meticulously compared to resolve the long-standing debate regarding the validity of Cricetinus Zdansky, 1928 and C. varians Zdansky, 1928. The findings indicate that the differences between the two are slight; as a result, C. varians can only be considered a chronosubspecies of T. triton, i.e., T. triton varians comb. nov., and Cricetinus should be recognized as a junior synonym of Tscherskia. We tentatively propose that, among the seven species once referred to Cricetinus, C. europaeus, C. gritzai, C. janossyi, and C. koufosi should be reassigned to Tscherskia, while C. beremendensis should be transferred to Allocricetus, and C. mesolophidos to Neocricetodon. Excluding Tscherskia sp. from the Late Pliocene Youhe fauna, there are no reliable Tscherskia fossils in China earlier than the Middle Pleistocene. Based on the current evidence, Tscherskia may have originated from Neocricetodon during the Early Pliocene in Europe and subsequently spread to Asia. T. triton is its sole surviving representative, which now exclusively inhabits East Asia.

Introduction

The late Middle Pleistocene Locality 2 of Shanyangzhai (village) has yielded abundant vertebrate fossils and one of the most common among them are the remains of hamsters—so far more than 50 skulls, 2,500 jaws, and very numerous isolated teeth have been discovered. These materials can be readily divided into two groups based on size. Xie & Li (2016) have described the small-sized group, recognizing two hamster species, Cricetulus longicaudatus and C. barabensis. The present article focuses on the remains of the large-sized group, providing a detailed morphological description and comparative study of these specimens.

The genus Cricetinus and its type species C. varians were erected by Zdansky (1928) on the basis of fossils from the renowned Middle Pleistocene Locality 1 of Zhoukoudian (i.e., the Peking Man Site), Beijing. Since then, hamster remains from several other Pleistocene mammal fossil sites have been continuously referred to C. varians, making it one of the most common micromammal species of Pleistocene faunas in northern China (refer to the synonymy of Tscherskia triton varians below). Kretzoi (1959) founded the second species of Cricetinus, C. europaeus, based on material from the Pliocene fauna of Csarnóta 2 in the Villány Mountains, southern Hungary. Following Kretzoi, five other hamster species have been successively allocated to Cricetinus (e.g., Hír, 1994; Wu & Flynn, 2017), rendering Cricetinus a widely distributed genus, both spatially and temporally. However, the validity of Cricetinus and C. varians has long been questioned by many researchers since the 1930s (e.g., Pei, 1936; Teilhard de Chardin & Pei, 1941; Teilhard de Chardin & Leroy, 1942; Gu, 1978; McKenna & Bell, 1997). The crux of this debate revolves around whether Cricetinus and C. varians are junior synonyms of Tscherskia and T. triton, respectively. In the present study, we examine the long-disputed issue of the validity of Cricetinus and C. varians based on a detailed description of large-sized hamster material from Locality 2 of Shanyangzhai, in order to enhance our understanding of the origin and evolution of extant Cricetinae taxa.

It is worth pointing out that, since Argyropulo’s (1933) work, Tscherskia has long been considered a subgenus of Cricetulus, with a few researchers maintaining this view until recently (e.g., Chen & Gao, 2000; Wang, Wu & Qiu, 2020). However, several molecular phylogenetic studies in recent years have demonstrated that Cricetulus in the traditional sense (usually including C. barabensis, C. longicaudatus, C. migratorius, C. kamensis, C. triton) is polyphyletic. As a result, the subgenera Tscherskia and Urocricetus (the Tibetan hamster) should be treated as two independent genera (e.g., Neumann et al., 2006; Steppan & Schenk, 2017; Lebedev et al., 2018; Ding et al., 2020; Romanenko et al., 2021). Furthermore, C. migratorius (the grey hamster) should also be recognized as an independent genus, and Lebedev et al. (2018) have proposed a new genus name, Nothocricetulus, for it. According to these studies, Cricetulus is generally more closely related to Nothocricetulus, Cricetus, and Allocricetulus when compared to Tscherskia. In addition to molecular phylogenetic evidence, morphological differences between T. triton and members of Cricetulus (in the traditional sense) are also apparent (Musser & Carleton, 2005) (Figs. S1–S3). For instance, T. triton has a considerably larger body size than the latter: the average body length of the former is approximately 157 mm, while the same measurement for the later ranges approximately from 90–100 mm (based on data provided by Chen & Gao, 2000). Moreover, we find that the third upper molar (M3) of T. triton bears an axioloph (sensu Freudenthal & Daams, 1988) (see “Discussion” for details), a feature that is, to the best of our knowledge, unique among all living Cricetinae species. The presence of the axioloph undoubtedly highlights the uniqueness of Tscherskia within Cricetinae, although its taxonomic significance warrants further investigation. For above reasons, we follow the prevailing opinion of researchers over the last two decades and no longer consider Tscherskia as a subgenus of Cricetulus, but rather as a distinct genus.

Geologic setting

Shanyangzhai Village (119°32′14.00″E, 40°5′17.82″N) is situated in the central part of Haigang District, Hebei Province, China, approximately 20 km from Bohai Bay. The Ordovician limestone of the Majiagou Formation south of the village reveals numerous fossil-bearing fissures or cave deposits due to quarrying activities. To date, four primary fossiliferous localities have been discovered, numbered Localities 1, 2, 3, and 4 of Shanyangzhai (abbreviated as Syz 1-4) in order from south to north (Fig. 1). The fossils from Syz 1 and Syz 3 primarily consist of macromammals, whereas Syz 2 and Syz 4 have yielded an abundance of micromammal fossils (Niu, Zhang & Fa, 2003; Kong, 2009; Wang et al., 2010; Zhang et al., 2010; Li & Zhang, 2011, 2013; Li, Zhang & Ao, 2013; Li, Zhang & Li, 2013a; Li, Zhang & Zheng, 2013b; Zhang & Li, 2015; Li, Li & Zhang, 2016; Xie & Li, 2016; Chen et al., 2021). Based on their mammalian components, these localities may have different ages. Although the majority of the original strata of Syz 2 have been disrupted by quarrying, the site, from which the hamster fossils studied in this article were obtained, is generally considered to date back to the late Middle Pleistocene (Zhang et al., 2010). This estimation mainly results from the overall faunal resemblance between Syz 2 and the Middle Pleistocene Locality 1 of Zhoukoudian (where the age of the primary fossiliferous deposits, i.e., layers 1–11, is generally considered to be ca. 0.6–0.2 Ma, Hu, 1985; Zhang, 2004; Chen & Zhou, 2009; Liu et al., 2014) and Jinniushan (ca. 0.31–0.2 Ma, Liu et al., 2014). We obtained a silty clay sample (field number 09SS1, laboratory number 11695) from about 2 m below the fossil-bearing horizon of Syz 2 and determined its absolute age using the electron spin resonance (ESR) technique at the State Key Laboratory of Earthquake Dynamics, Institute of Geology of China Earthquake Administration. The test result demonstrates an age of 300 ± 30 ka for the sample, implying that the fossil-bearing horizon located above the sampling location should be somewhat younger than this age. Kong (2009) dated the fauna between (1.8 ± 0.2) × 105 years and (2.0 ± 0.2) × 105 years ago using the thermo-luminescence (TL) technique, but the precise sampling layers in Kong (2009) require further verification. Based on the aforementioned absolute age dating results and faunal comparisons, we tentatively date Syz 2 to the late Middle Pleistocene, approximately 0.2 Ma.

Figure 1 Geographic locations of Syz 1~4.

Satellite photo credit: Google Earth. © CNES/Airbus.

Material, methods, and abbreviation

Material

The hamster fossils from Syz 2 examined in this study are housed in the Department of Geology, Northwest University (Xi’an, Shaanxi Province, China). For comparative purposes, we observed about 50 skull specimens (including mandibles) of extant Tscherskia triton (the greater long-tailed hamster), all of which were collected from Shaanxi Province and stored in the College of Life Sciences, Northwest University. Based on the collection sites indicated on the labels of these specimens, they might belong to the subspecies T. triton incanus and T. triton collinus (and possibly T. triton triton and T. triton fuscipes), following the subspecies classification and geographical distribution of T. triton summarized by Chen & Gao (2000).

Methods

The skull morphological terminology primarily follows Wang & Qiu (2018) and Voss (1988). For the description of molar morphology, we mainly follow Freudenthal & Daams (1988), Freudenthal, Hugueney & Moissenet (1994), and Li et al. (2018). Anatomical abbreviations for upper molars are M1, M2, and M3, while lower molars are represented by m1, m2, and m3. Measurements of the skull and mandible mainly follow the methodology described by Yang et al. (2005) and Xia et al. (2006). For molar measurements, L and W denote the maximum length and width, respectively. All measurements were obtained using a ZEISS Smartzoom5 automated digital microscope. Some images were reversed for ease of comparison and are indicated by an underlined label. Annotations of the synonymy list are in accordance with Matthews (1973). This work and the nomenclatural act it contains have been registered in ZooBank (https://zoobank.org/urn:lsid:zoobank.org:act:833CA9EC-1051-4C8E-A032-04DF2AC46994).

Abbreviation

NWU, Northwest University, Xi’an; IVPP, Institute of Vertebrate Paleontology and Paleoanthropology, Beijing; Syz 1-4, Locality 1, 2, 3, 4 of Shanyangzhai; ZKD, Zhoukoudian (=Choukoutien); JNS, Jinniushan; RZD, Renzidong.

Results

Systematic paleontology

Mammalia Linnaeus, 1758

Rodentia Bowdich, 1821

Cricetidae Rochebrune, 1883

Cricetinae Fisher, 1817

Tscherskia Ognev, 1914

1928 Cansumys, Allen, 1928

1928 Cricetinus, Zdansky, 1928

1929 Asiocricetus, Kishida, 1929

Type species: Tscherskia albipes Ognev, 1914 (=Cricetus (Cricetulus) triton de Winton, 1899).

Referred species: T. europaeus (Kretzoi, 1959); T. rusa (Storch, 1974) ?; T. gritzai (Topachevsky & Skorik, 1992); T. janossyi (Hír, 1996); T. koufosi (Koliadimou, 1996).

Geographic distribution and geologic age: Southeastern Europe, Early Pliocene (MN 15, ca. 5–3.5 Ma) to early Middle Pleistocene (ca. 0.7 Ma); Southwestern Asia, Holocene ?; northern China, transitional region between northern and southern China, except T. sp. from Youhe fauna with an age of Late Pliocene (ca. 3.40–2.59 Ma) (Yue & Xue, 1996; Xie, Zhang & Li, 2021), all other credible material of Tscherskia with an age not earlier than Middle Pleistocene.

Diagnosis: Medium-sized cricetids typically between Cricetulus and Cricetus; molars brachyodont; mesolophes of M1-3 usually present, either free or connected to the metacone, but rarely reaching the buccal tooth edge; M3 with an anteroposteriorly directed axioloph rather than an anterolaterally extended protolophule II; anteroconids of m1 either divided or undivided; mesolophids on m3 almost always present and well-developed; mesolophids also often present on m1 and m2 but rarely reaching the lingual tooth edge (modified from Xie, Zhang & Li, 2021).

Remarks: Hír (1996a, 1997) once proposed the diagnosis of Cricetinus (i.e., Tscherskia) as follows: “the undivided anteroconid on the m1 molars with a smooth and convex oral surface; the mesolophids missing or short on the m1-m2 molars; M1-M2 crowns characterized by the missing or weekly developed mesolophes; the posterior metalophule rare on M2.” Our observation demonstrates that the diagnosis proposed by Hír is not comprehensive and warrants further revision. Therefore, the diagnosis of Tscherskia is redefined here as stated above.

Tscherskia triton (de Winton, 1899)

Tscherskia triton varians comb. nov. (Zdansky, 1928)

p1927 Cricetulus cfr. songarus Pallas—Young, 1927, p.24

*1928 Cricetinus varians gen. et sp. nov.—Zdansky, 1928, p.54

.1930 Cricetinus varians Zdansky—Schaub, 1930, p.37

1931 Cricetinus varians Zdansky—Pei, 1931, p.12

1932 Cricetinus varians Zdansky—Young, 1932, p.4

.1934 Cricetinus varians Zdansky—Schaub, 1934, p.30

1934 Cricetinus varians Zdansky—Young, 1934, p.58

p1936 cf. Cricetinus varians Zdansky—Teilhard de Chardin, 1936, p.16

1936 Cricetinus varians Zdansky—Pei, 1936, p.59

1939 Cricetinus varians Zdansky—Pei, 1939, p.153

p?1940 Cricetinus (Cricetulus) varians Zdansky—Pei, 1940, p.42

p1941 Cricetulus varians (Zdansky)—Teilhard de Chardin & Pei, 1941, p.49

p1942 Cricetulus (Cricetinus) varians (Zdansky)—Teilhard de Chardin & Leroy, 1942, p.35, p.93

1977 Cricetulus varians (Zdansky)—Gai & Wei, 1977, p.290

1978 Cricetulus triton (Zdansky)—Gu, 1978, p.164

1980 Cricetulus varians (Zdansky)—Zhang, Zou & Zhang, 1980, p.156

1983 Cricetulus varians (Zdansky)—Zheng, 1983, p.231

1984 Cricetinus varians Zdansky—Zheng, 1984a, p.185

1985 Cricetulus varians (Zdansky)—Zhang et al., 1985, p.73

1985 Cricetinus varians Zdansky—Zheng et al., 1985, p.117

1986 Cricetulus varians (Zdansky)—Zhang, Wei & Xu, 1986, p.36

1990 Cricetulus triton (de Winton)—Sun & Jin, 1990, p.35

1993 Cricetinus varians Zdansky—Zheng & Han, 1993, p.65

p?1996 Cricetinus varians Zdansky—Cheng et al., 1996, p.38

2002 Cricetinus varians Zdansky—Jin, 2002, p.95

2004 Cricetinus varians Zdansky—Jin et al., 2004, p.284

2004 Cricetulus triton (de Winton)—Tong et al., 2004, p.855

?2009 Cricetinus varians Zdansky—Jin et al. (2009), p.177

2010 Cricetinus varians Zdansky—Zhang et al., 2010, p.73

2015 Tscherskia triton (de Winton)—Liu et al., 2015, p.610

2017 Tscherskia triton (de Winton)—Chen et al., 2017, p.847

2018 Cricetulus varians (Zdansky)—Tong et al., 2018, p.287

2018 Cricetinus varians Zdansky—Wu et al., 2018, p.1396

2020 Cricetulus varians (Zdansky)—Wang, Wu & Qiu, 2020, p.104

2021 Cricetulus varians (Zdansky)—Huang et al., 2021, p.269

2021 Cricetulus triton (de Winton)—Huang et al., 2021, p.269

Lectotype: As previously mentioned, Zdansky (1928) established Cricetinus and Cricetinus varians based on large-sized hamster material from Locality 1 of Zhoukoudian, which included eight maxillary fragments, nine larger and some smaller mandibular fragments, 1 isolated M1, and 3 isolated m1s. However, Zdansky did not designate a holotype for the new genus and its type species, so all these specimens should be considered the syntypes. Wang, Wu & Qiu (2020, pp.104–105) selected IVPP RV 340020 (original catalogue number C/C. 1049), an anterior portion of the skull with right M1-3 and left M1-2 figured by Young (1934, Text-fig. 19, 1, 1a, 1b; Pl. 5, Fig. 9) and Zheng (1984a, Fig. 1, C), as the lectotype of C. varians. However, this designation should be considered invalid according to ICZN (1999, Art. 74.2), because IVPP RV 340020 does not belong to the syntypes, although it was also collected from Locality 1, possibly even from the same layer as the syntypes (Young, 1934, p.63). Therefore, the fragmentary right upper jaw with M1-3 figured by Zdansky (1928, Taf. 5, Fig. 4) is here designated as the lectotype for Tscherskia triton varians (Lagrelius Collection housed in the Museum of Evolution, Uppsala University, Sweden), and other specimens in the type series should be considered the paralectotypes. The paralectotypes listed by Wang, Wu & Qiu (2020, p.104) are also invalid for the same reason discussed for the lectotype and should only be viewed as referred specimens.

Type locality and geologic age: Locality 1 of Zhoukoudian, Beijing. The deposits of Locality 1, also known as Zhoukoudian Formation, are about 40 m in thickness and traditionally divided into 1 to 13 layers from top to bottom, representing a period from approximately 0.78 to 0.2 Ma. This division scheme was published by Jia (1959), who adopted a similar scheme first proposed by Teilhard de Chardin & Young (1929), and has been widely followed since. Zdansky (1923, p.86) also published two profiles of deposits of Locality 1 (called Loc. 53 by Zdansky) from which the type specimens of C. varians and other fossils studied by Zdansky (1928) were collected. Teilhard de Chardin & Young (1929, p.179, footnote) considered that the sections given by Zdansky (1923, p.86) correspond probably to some part of their layers 5 and 6, although they also stated that a precise correlation with Zdansky’s (1923) profile was rather difficult to establish. If Teilhard de Chardin and Young are correct, according to Xu et al. (1997, p.219, Table 1), their layers 5 and 6 should essentially correspond to layers 4 to 6 of Jia’s (1959) scheme, which cover a period of approximately 0.3–0.4 Ma in the Middle Pleistocene (Chen & Zhou, 2009, Table 1).

Table 1 Measurements and comparisons of skulls and mandibles of Tscherskia triton varians from Syz 2 and extant T. triton (mm)#.

	T. triton varians of Syz 2	The extant T. triton	
N	Min.	Mean	Max.	SD	CV	N	Min.	Mean	Max.	SD	CV	
Palatal length	4	16.39	18.01	20.04	1.57	8.7%	40	13.72	17.21	20.13	1.70	9.9%	
Upper diastema length	13	9.19	10.83	12.00	0.86	7.9%	46	7.64	9.98	12.31	1.15	11.5%	
Length of the incisive foramen	11	6.26	7.11	7.93	0.51	7.2%	46	4.81	6.35	7.66	0.76	12.0%	
Anterior palatal breadth	21	3.14	3.60	4.06	0.27	7.4%	44	2.39	3.18	3.78	0.31	9.8%	
Posterior palatal breadth	11	3.34	3.78	4.02	0.18	4.7%	39	2.48	3.21	4.02	0.29	9.1%	
Width of nasal*	6	1.88	2.18	2.34	0.16	7.4%	45	1.95	2.44	3.31	0.29	12.0%	
Frontal suture length	1		9.66				43	8.83	10.68	12.62	0.95	8.9%	
Parietal suture length	1		5.78				41	5.33	6.38	7.49	0.51	8.0%	
Interparietal length	1		5.83				42	2.70	3.88	6.16	0.58	15.0%	
Interparietal width	1		10.79				40	7.70	9.11	10.47	0.68	7.5%	
Lower diastema length	21	4.89	5.76	6.70	0.56	9.7%	38	4.77	5.70	6.67	0.46	8.0%	
Depth of mandible under anterior edge of alveolus	65	3.54	4.48	6.52	0.50	11.2%	40	3.55	4.56	5.91	0.52	11.4%	
Depth of mandible between two roots of m1	78	4.31	5.27	6.63	0.48	9.2%	40	3.60	4.91	6.59	0.71	14.4%	
Depth of mandible between two roots of m2	142	3.84	4.80	6.05	0.45	9.4%	40	3.04	4.36	6.05	0.68	15.7%	
Depth of mandible between two roots of m3	162	2.83	3.87	4.98	0.44	11.3%	37	2.74	3.57	5.10	0.59	16.4%	
Depth of mandible under posterior edge of alveolus	187	2.42	3.33	4.21	0.32	9.6%	40	2.40	3.17	4.44	0.42	13.2%	
Length of mandible from the condyle	2	20.38	22.24	24.09	1.86	8.3%	38	16.05	20.30	24.99	2.07	10.2%	
Distance from coronion to gonion ventrale	1		13.13				29	7.58	10.23	12.95	1.36	13.3%	
Notes:

* “Width of nasal” here indicates the distance between the two junctions of the nasal, premaxilla and frontal.

#Refer to Datasets S1–S4 for raw data.

Geographic distribution and geologic age: Northern China, transitional region between northern and southern China, late Early Pleistocene to Late Pleistocene.

Referred specimens from Syz 2: 21 incomplete skulls (NWUV 1489.a1-21); 10 maxillae with bilateral toothrows (NWUV 1489.b1-10); 73 left maxillae (NWUV 1489.c1-73); 74 right maxillae (NWUV 1489.d1-74); 185 left mandibles (NWUV 1489.e1-185); 215 right mandibles (NWUV 1489.f1-215); 3 mandibles with bilateral branches (NWUV 1489.g1-3); 55 left M1s (NWUV 1489.h1-55); 54 right M1s (NWUV 1489.i1-54); 46 left M2s (NWUV 1489.j1-46); 35 right M2s (NWUV 1489.k1-35); 2 left M3s (NWUV 1489. l1-2); 8 right M3s (NWUV 1489.m1-8); 16 left m1s (NWUV 1489.n1-16); 22 right m1s (NWUV 1489.o1-22);15 left m2s (NWUV 1489.p1-15); 19 right m2s (NWUV 1489.q1-19); 7 left m3s (NWUV 1489.r1-7); 8 right m3s (NWUV 1489.s1-8).

Measurements: Refer to Tables 1 and 2, Datasets S1, S3, S5 and S7.

Table 2 Measurements and comparisons of molars of Tscherskia triton and “Cricetinus” varians (mm)#.

	M1-3	M1	M2	M3	m1-3	m1	m2	m3	
L	L	W	L	W	L	W	L	L	W	L	W	L	W	
T.
triton
varians
from
Syz 2	N	34	83	84	83	84	47	46	56	89	105	106	107	74	73	
Min.	5.05	2.12	1.40	1.69	1.44	1.21	1.27	5.40	1.97	1.19	1.65	1.39	1.65	1.27	
Mean	5.44	2.33	1.56	1.85	1.56	1.43	1.40	5.67	2.14	1.31	1.81	1.54	1.78	1.43	
Max.	5.69	2.49	1.72	2.00	1.78	1.55	1.48	5.98	2.28	1.43	1.96	1.70	1.93	1.56	
SD	0.15	0.07	0.06	0.06	0.06	0.07	0.04	0.13	0.06	0.04	0.06	0.05	0.06	0.05	
CV	2.8%	3.2%	4.0%	3.4%	3.9%	5.2%	3.1%	2.4%	2.6%	3.4%	3.4%	3.2%	3.4%	3.6%	
C.
varians
from ZKD*	N	12	21	20	18	18	14	14	49	56	56	57	57	51	50	
Min.	5.25	2.15	1.45	1.65	1.45	1.35	1.25	4.70	1.90	1.15	1.60	1.30	1.60	1.20	
Mean	5.60	2.32	1.56	1.80	1.59	1.44	1.41	5.52	2.06	1.30	1.72	1.43	1.71	1.36	
Max.	5.85	2.50	1.70	1.90	1.65	1.50	1.50	5.85	2.25	1.40	1.90	1.55	1.90	1.55	
SD	0.19	0.10	0.07	0.06	0.07	0.05	0.06	0.20	0.08	0.06	0.09	0.05	0.09	0.06	
CV	0.4%	4.3%	4.6%	3.5%	4.2%	3.3%	4.1%	3.6%	3.9%	4.5%	5.1%	3.7%	5.0%	4.7%	
C.
varians
from JNS**	N	5	9	9	9	9	5	5	12	20	20	18	18	12	12	
Min.	5.28	2.18	1.40	1.77	1.44	1.42	1.30	5.33	1.96	1.20	1.70	1.36	1.67	1.30	
Mean	5.42	2.34	1.46	1.95	1.52	1.49	1.38	5.59	2.07	1.29	1.73	1.45	1.79	1.35	
Max.	5.60	2.46	1.51	2.00	1.60	1.57	1.50	6.00	2.27	1.38	2.00	1.60	2.00	1.47	
C.
varians
from RZD***	N		35	35	25	25	1	1	2	52	52	50	50	18	2	
Min.		2.0	1.2	1.5	1.25			4.95	1.85	1.1	1.45	1.15	1.4	1.15	
Mean		2.15	1.31	1.64	1.32	1.7	1.5	4.98	2.01	1.14	1.59	1.3	1.59	1.26	
Max.		2.3	1.4	1.8	1.4			5	2.2	1.25	1.65	1.4	1.7	1.3	
The extant
T.
triton	N	42	47	47	47	47	42	42	36	39	38	39	39	36	36	
Min.	5.01	2.18	1.45	1.64	1.47	1.30	1.26	5.26	1.95	1.24	1.68	1.41	1.66	1.32	
Mean	5.36	2.32	1.55	1.82	1.59	1.43	1.40	5.58	2.16	1.32	1.81	1.51	1.78	1.40	
Max.	5.79	2.52	1.68	2.00	1.75	1.67	1.62	5.86	2.38	1.41	1.94	1.63	1.99	1.58	
SD	0.15	0.07	0.06	0.08	0.05	0.08	0.06	0.15	0.09	0.04	0.06	0.05	0.07	0.06	
CV	2.7%	3.1%	3.7%	4.2%	3.4%	5.4%	4.1%	2.8%	4.0%	3.1%	3.4%	3.5%	4.0%	4.3%	
Notes:

* Quoted from Zheng (1984a).

** Quoted from Zheng & Han (1993).

*** Quoted from Jin et al. (2009).

#Refer to Datasets S5–S8 for raw data.

Diagnosis: Tscherskia triton varians is highly similar to extant T. triton in size and most of the molar characters (see “Discussion”). However, the former exhibits slightly higher frequencies of mesolophids on m1 and m2 (refer to Table 7). In the majority of skull and mandible measurements, the mean values for T. t. varians may be slightly larger than those of extant T. triton.

Table 7 Comparisons of frequencies of mesolophids on lower molars among the species of Cricetinus, Tscherskia, Cricetulus, Nothocricetulus, and Allocricetus.

Species	Localities	Geologic age	Frequencies of mesolophids on m1s	Frequencies of mesolophids on m2s	Frequencies of mesolophids on m3s	Sources	
Present paper	Original references	
Tscherskia triton (type species)	/	Shaanxi Province, China	Recent	30% (21/69)	87% (60/69)	100% (63/63)	Present paper	
T. triton varians	/	Syz 2, Hebei Province, China	Late Middle Pleistocene	43% (44/103)	95% (162/170)	100% (134/134)	Present paper	
Cricetinus varians	ZKD Loc. 3, Beijing, China	Late Middle Pleistocene or Late Pleistocene	61% (54/89)	97% (86/89)	100% (89/89)	Zheng (1984a)	
Jinniushan, Liaoning Province, China	Late Middle Pleistocene	67%	91% (20/22)	100%	Zheng & Han (1993)	
ZKD Loc. 1 (type locality), Beijing, China	Middle Pleistocene	70% (40/57)	93% (53/57)	100% (57/57)	Zheng (1984a)	
T. triton varians ?	Cricetinus varians	Renzidong, Anhui Province, China	Early Early Pleistocene	present	present	present	Jin et al. (2009)	
T. europaeus	Cricetinus europaeus	Csarnóta 2 (type locality), Hungary	Pliocene	33.3% (2/6)	71.4% (5/7)	100% (5/5)	Hír (1994)	
T. gritzai	Cricetinus gritzai	Odessa (type locality), Ukraine	Pliocene	present	present	present	Topachevsky & Skorik (1992)	
T. janossyi	Cricetinus janossyi	Osztramos 7 (type locality) and Csarnóta 2, Hungary	Pliocene	38.9% (7/18)	95% (19/20)	100% (15/15)	Hír (1996b)	
T. koufosi	Cricetinus koufosi	Mygdonia basin (type locality), Greece	Early Pleistocene	0	–	–	Koufos et al. (2001)	
Neocricetodon mesolophidos	Cricetinus mesolophidos	Yushe basin (type locality), Shanxi Province, China	Pliocene	100%	100%	perhaps 100%	Wu & Flynn (2017)	
Cricetulus barabensis (type species)	/	Shaanxi Province, China	Recent	0 (0/8)	0 (0/8)	0 (0/8)	Present paper	
C. longicaudatus	/	Shaanxi Province, China	Recent	0 (0/23)	0 (0/23)	26.1% (6/23)	Present paper	
Nothocricetulus migratorius (type species)	Cricetulus migratorius	Krak des Chevaliers, Syria	Recent	0?	0?	very often	Pradel (1981)	
Meydan, Toros Mountains, Turkey	Holocene	0?	10%	81%	Hír (1993a)	
Allocricetus bursae (type species)	/	Tarko Rockshelter 1, Hungary	Early Middle Pleistocene	0?	10%	85%	Hír (1993a)	
Tarko Rockshelter 2–10, Hungary	0?	2%	60%	
Tarko Rockshelter 11–12, Hungary	0?	16%	84%	
Tarko Rockshelter 13–15, Hungary	0?	28%	100%	
Tarko Rockshelter 16–18, Hungary	0?	33%	93%	
A. ehiki	/	Villány 3 and Esztramos 3, Hungary	Early Pleistocene	0?	52%	91%	Hír (1993a, 1993b)	
ZKD Loc. 12, 18, Beijing, China	Early Pleistocene	5% or 0?	4%	100% (47/47)	Zheng (1984a)	
A. beremendensis	Cricetinus beremendensis	Beremend 15 (type locality) and Csarnóta 4, Hungary	Pliocene	0% (0/72)	14.8% (9/61)	100% (53/53)	Hír (1994)	

Remarks: The minor differences between T. triton varians and extant T. triton can only be observed when a statistically significant number of specimens are available. The reason for referring all items listed in the synonymy, most of which have limited material, to T. triton varians is solely based on their geologic age. Thus, this should be considered a temporary expedient.

Description

(1) Skull (Fig. 2)

Figure 2 Skulls of Tscherskia triton varians from Syz 2.

(A), (E), (H), (K), NWUV 1489.a8, incomplete skull; (B), (F), (I), NWUV 1489.a21, incomplete skull; (C), (G), (J), NWUV 1489.a6, incomplete skull; (D) NWUV 1489.a7, incomplete skull. (A–D), dorsal view; (E–G), ventral view; (H–J), lateral view; (K), anterior view. The underlined label indicates the image has been reversed. Abbreviations: arza, anterior root of the zygomatic arc; aui, alveolus of the upper incisor; F, frontal; fc, frontal crest; inf, incisive foramen; iof, infraorbital foramen; Ip, interparietal; M, maxilla; mt, masseteric tubercle; N, nasal; P, parietal; pbhp, posterior border of the hard palate; Pm, premaxilla; ppf, posterior palatine foramen; sof, supraorbital foramen; zp, zygomatic plate.

The skull description primarily relies on the relatively well-preserved specimen NWUV 1489.a8, with reference to other specimens.

Dorsal view: The nasal exhibits a narrow posterior and a wide anterior aspect. At its junction with the frontal, it is narrowest, then gradually widening anteriorly, and slightly narrowing again at the anterior border. The anterior-most point of the orbit is slightly anterior to the transverse level of the posterior end of the nasal. NWUV 1489.a7 has a larger skull width than normal due to post-mortem deformation, but it retains the complete interparietal, which is approximately pentagonal in shape (Fig. 2D), resembling that of extant T. triton. The frontal crest is more pronounced in adults, particularly in elderly individuals, extending posteriorly from the upper edge of the orbit, beyond the parietal bone, and reaching at least the anterior border of the interparietal bone.

Lateral view: The upper contour of the skull presents a gentle arc, but this shape is often lost due to post-mortem deformation. The alveolus of the upper incisor creates a well-defined semicircular crest on the lateral surfaces of the premaxilla and maxilla. The upper portion of the infraorbital foramen is fan-shaped, while its lower portion is slit-like. The outer wall of the zygomatic plate is slightly concave. Both the anterior and posterior edges of the zygomatic plate exhibit a gentle arc shape; the former is slightly convex anterodorsally, and the latter is slightly concave anterodorsally, with the two edges nearly parallel. The anterior root of the zygomatic arch is weak, measuring about 2–3 times narrower the width of the zygomatic plate. The small supraorbital foramen is situated posterior to the interorbital constriction and just below the supraorbital margin.

Ventral view: The incisive foramen is elongated and narrow, with an obvious distance separating its posterior edge from the anterior edge of M1. The premaxillary-maxillary suture traverses the incisive foramen at about the anterior 2/5 of the foramen. The anterior-most point of the zygomatic plate approximately aligns with the center of the incisive foramen in the mediolateral direction. The masseteric tubercle is positioned at the base of the zygomatic plate, exhibiting a rough surface. Two posterior palatine foramina are almost situated on the connecting line of the posterior roots of the two M2s. The posterior border of the hard palate extends slightly beyond the posterior edge of M3 or is flush with it. The two molar series are not completely parallel, but slightly divergent anteriorly.

(2) Mandible (Fig. 3)

Figure 3 Mandibles of Tscherskia triton varians from Syz 2.

(A–B) NWUV 1489.f206, right mandible; (C–E) NWUV 1489.f207, right mandible; (F–H) NWUV 1489.e169, left mandible; (I–J) NWUV 1489.e164, left mandible. (A), (C), (G), (I), buccal view; (B), (D), (H), (J), lingual view; (E–F), occlusal view. Abbreviations: ap, angular process; cdp, condyloid process; crp, coronoid process; fmg, foramen in the middle of the groove (g); g, groove between the alveolus of molars and the base of the coronoid process; i2b, bulge formed by i2; mdf, mandibular foramen; mn, mandibular notch; mr, masseteric ridge; mstf, masseteric fossa; mtf, mental foramen.

The lower edge of the mandible extends anteriorly in an arc from the base of the angular process. The mental foramen is small and round, located anteroventral to the anterior root of m1. The masseteric ridge is thin yet clearly evident, ending beneath m1 and posterodorsal to the mental foramen. The coronoid process is slender and hook-shaped, extending posterodorsally. A noticeable bulge formed by the posterior end of the lower incisor is present at the base of the condylar process, situated anteroventral to the mandibular notch on the buccal side of the mandible. The angular process extends in a posteroventral direction. The mandibular notch extends slightly further anteriorly than the notch between the condylar process and the angular process, with the latter slightly wider than the former. The mandibular foramen is oval and located at the base of the condylar process. The groove between the alveolus of toothrow and the base of the coronoid process slopes gently in the posterior direction, but not as steep as that of murines; a small foramen of unclear function is situated in the middle of the groove. On the inner side of the mandible, numerous small nutrient foramina are typically found in the area beneath the molar series.

Measurements of skulls and mandibles are provided in Table 1 and Datasets S1 and S3.

(3) Teeth (Figs. 4, 5)

Figure 4 Left upper molars of Tscherskia triton varians from Syz 2.

(A) NWUV 1489.a5; (B) NWUV 1489.a14; (C) NWUV 1489.a21; (D) NWUV 1489.b1; (E) NWUV 1489.c3; (F) NWUV 1489.c5; (G) NWUV 1489.c16. The arrow indicates the axioloph.

Figure 5 Right lower molars of Tscherskia triton varians from Syz 2.

(A) NWUV 1489.f8; (B) NWUV 1489.f13; (C) NWUV 1489.f22; (D) NWUV 1489.f28; (E) NWUV 1489.f31; (F) NWUV 1489.f49; (G) NWUV 1489.f56.

I2: The anterior end of the upper incisor (I2) points ventrally, and its posterior end terminates in an anteroventral position relative to the infraorbital foramen. The enamel layer covers the entire labial surface of I2, which is smooth and devoid of ridges, and also extends to cover a small portion of the lateral surface.

M1: The M1 is kidney-shaped, with an obtuse anterior edge, a comparatively straight buccal edge (but with a noticeable outward protrusion at the metacone), and an arc-shaped lingual edge. The degree of alternating of the opposite main cusps on M1 is small, as is the case for M2 and M3. The anterocone is broad and always splits posteriorly into two equal-sized cusps. In some specimens, the anterocone also exhibits a certain degree of separation from the mesial surface, and in a few cases, this separation is even pronounced. The lingual anterolophule is invariably present, while the buccal anterolophule is observed in 89.1% (41/46) of specimens. A small number of specimens (3.9%, 6/154) exhibit the spur of the anterolophule, which is thin and weak, with five instances extending to the buccal margin of the tooth (Fig. 4B). The presence of the protolophule I is detected in 57.4% (27/47) of specimens. The protolophule II is relatively weak, and even absent in a few specimens. The loph connecting the anterior arm of the hypocone and the metacone, in our opinion, should be viewed as the mesoloph, as in most specimens, there is an obvious contact trace between the loph and the metacone, implying that the loph does not originate from the metacone. In a few specimens, however, this loph can be completely fused with the metacone without any trace, making it difficult to determine whether the metalophule I contributes to the formation of the loph in these cases. No specimens exhibit a mesoloph with a free end. The metalophule II is present but weakly developed in most specimens. The posterosinus is small and shallow, with only a vestige observable in specimens exhibiting severe abrasion. The tooth has four roots.

M2: The M2 is approximately square in shape. The buccal anteroloph is more developed than its lingual counterpart, with the latter occasionally nearly absent. The position of the buccal anteroloph is also elevated compared to the lingual one. The protolophule is double. The mesoloph resembles that of M1 but is relatively thicker. It may either merge with the metacone or display an evident contact trace between them, yet it never has a free end. In some specimens, the mesoloph extends to the tooth edge by adhering to the anterior wall of the metacone (Figs. 4B and 4F). The metalophule II is consistently present, albeit comparatively weak. The posterosinus is also small. The tooth is four-rooted.

M3: The posterior portion of M3 is notably reduced, with both the hypocone and metacone significantly smaller than those of M1 and M2. This results in the occlusal outline of M3 resembling a relatively obtuse equilateral triangle. The buccal anteroloph is more developed and positioned higher than the lingual counterpart, which is either absent or extremely weak. The protolophule I is consistently present. The most notable character of M3 is the presence of the axioloph, which originates from the junction of the protolophule I and the anterior arm of the protocone, and extends posteriorly. A small groove forms between the axioloph and paracone. Occasionally, the central part of the groove closes due to the proximity or fusion of the axioloph and paracone, leading to the formation of a small pit in the upper portion of the groove (Figs. 4D and 4F). The morphology of the mesoloph is similar to that of M1 and M2. The metalophule II and posterosinus are absent. In some specimens, the mid-segment of the posteroloph (or the posterior arm of the hypocone) inflates into a small cusp, situated between the hypocone and metacone (Figs. 4C and 4G). The tooth possesses three roots.

In a very small number of specimens, the upper molars exhibit morphological variation in certain structures. For example, the protolophule II on M2 occasionally assumes a form similar to that on M3, and vice versa.

i2: The anterior part of the lower incisor (i2) extends anterodorsally, and the posterior end of it terminates at the base of the condylar process, forming a prominent bulge on the buccal side of the mandible. The enamel layer covers the whole labial surface, which is smooth and devoid of ridges, as well as about half of the lateral surface.

m1: The occlusal outline of m1 is comparatively elongated and gradually narrows from posterior to anterior. The anteroconid is bisected into two approximately equal-sized cusps in most specimens (93.0%, 80/86). In these specimens, the vast majority of anteroconids are slightly bifid posteriorly, although specimens with a more pronounced degree of posterior separation are occasionally observed. From an anterior perspective, the anteroconid is either weakly divided (in young individuals) or undivided (in middle-aged and elderly individuals). A small proportion of specimens (7.0%, 6/86) possess anteroconids split into three small cusps (Fig. 5E). Undivided anteroconids are observed only in heavily worn specimens. In the vast majority of specimens (97.6%, 82/84), the anterolophulid is single and connects either to the buccal anteroconulid (70.7%, 58/82), the midpoint between the two anteroconulids (26.8%, 22/82), or the lingual anteroconulid (2.4%, 2/82). In a very few specimens (2.4%, 2/84), the anterolophulid possesses two branches that connect to the two anteroconulids respectively. The bottom of the anterosinusid is significantly higher than that of the protosinusid. In 43% of the specimens (44/103) (Table 3), a mesolophid is present, which is consistently low, short, and weak. The mesolophid either connects to the metaconid (18.2%, 8/44) or has a free end (81.8%, 36/44). In the latter case, the longest free-ended mesolophid does not exceed half the distance from the base to the lingual tooth edge, and in most cases, it only presents as a spine-like projection. The transitional part from the hypoconid to posterolophid is generally slender, but subsequently the posterolophid rapidly swells into a well-defined small cusp. The posterolophid often does not continue anteriorly to connect with the entoconid, resulting in an open posterosinusid in most cases. The cingulum commonly presents at the entrances of the protosinusid and sinusid, occasionally forming a small but distinct ectostylid at the entrance of the latter. The tooth has two roots.

Table 3 Comparisons of mesolophids of m1s between Tscherskia triton and “Cricetinus” varians.

Species and localities	Frequencies of mesolophids on m1	
C. varians of ZKD Loc. 1	70% (40/57)	
C. varians of JNS	67%	
C. varians of ZKD Loc. 3	61% (54/89)	
T. triton varians of Syz 2	43% (44/103)	
The extant T. triton	30% (21/69)	

m2: The occlusal outline of m2 exhibits a rounded square shape, with a width greater than that of m1 and m3. The lingual anterolophid is weakly developed or absent, while the buccal anterolophid is always well developed. In 95.2% of specimens (158/166) the mesolophid is present, exhibiting various morphologies that can be essentially categorized into four types (Table 4): I. having a free end; II. connected to the metaconid; III. reaching the lingual tooth edge (10.2%, 16/157) (Fig. 5A); and IV. connected to the entoconid (2.5%, 4/157). Within these morphotypes, I and II are present in most specimens, but the boundaries between the two are sometimes difficult to distinguish. The length of the mesolophid also varies, but most do not exceed 1/2 of the distance from the base to lingual tooth edge. The morphology of the posterolophid and the development of the cingulum are similar to those on m1, except that the lingual edge of the mesosinusid of m2 also occasionally bears the cingulum. The tooth has two roots.

Table 4 Comparisons of mesolophids of m2s between Tscherskia triton and “Cricetinus” varians.

Species and localities	Frequencies of mesolophids on m2	Proportions of each morphotype of mesolophids on m2	
I or II*	III*	IV*	
C. varians of ZKD Loc. 1	93% (53/57)	—	11.3% (6/53)	—	
C. varians of JNS	91% (20/22)	—	0 (0/20)	—	
C. varians of ZKD Loc. 3	97% (86/89)	—	14.0% (12/86)	—	
T. triton varians of Syz 2	95% (158/166)	87.3% (137/157)	10.2% (16/157)	2.5% (4/157)	
The extant T. triton	87% (60/69)	96.6% (58/60)	1.7% (1/60)	1.7% (1/60)	
Note:

* I, having a free end; II, connected to the metaconid; III, reaching the lingual tooth edge; IV, connected to the entoconid.

m3: The posterior part of m3 is generally contracted, though a small number of specimens exhibit no obvious contraction (Fig. 5G). In most specimens, the entoconid is significantly reduced compared to that of m1 and m2, while the hypoconid often experiences only slight reduction. Similar to m2, the lingual anterolophid of m3 is also weakly developed and the buccal one is comparatively more pronounced; however, the lingual anterolophid is present in nearly all m3 specimens. The mesolophid is present in all but one specimen (99.2%, 129/130), and its morphology varies, falling into five types (Table 5): I. unbranched (59.4%, 76/128), connected to the lingual tooth edge (Figs. 5A, 5C, 5E and 5G); II. bifurcated (35.2%, 45/128), with one branch connected to the lingual tooth edge and the other to the metaconid (Figs. 5B and 5F); III. trifurcated (0.8%, 1/128), with branches connected to the lingual tooth edge, metaconid, and junction of the hypoconid and entoconid, respectively; IV. unbranched (3.9%, 5/128), connected to the metaconid (Fig. 5D); and V. having a free end (0.8%, 1/128). The posterolophid is somewhat different from that of m1 and m2, primarily in that it usually merges with the entoconid to close the posterosinusid. The posterolophid also exhibits some degree of swelling and appears as a cusp when subjected to slight wear, resulting in three side-by-side cusps on the posterior part of m3. The cingulum is usually absent at the entrance of the sinusid but is often more developed at the entrance of the mesosinusid, occasionally merging with the end of the mesolophid to form a small cusp. The tooth possesses two roots.

Table 5 Comparisons of mesolophids of m3s between Tscherskia triton and “Cricetinus” varians.

Species and localities	Frequencies of mesolophids on m3	Proportions of each morphotype of mesolophids on m3	
I*	II*	III*	IV*	V*	
C. varians of ZKD Loc. 1	100% (57/57)	71%**	—	—	
C. varians of JNS	100%	100%**	—	—	
C. varians of ZKD Loc. 3	100% (89/89)	91%**	—	—	
T. triton varians of Syz 2	99.2% (129/130)	95.3% (122/128)**	3.9%
(5/128)	0.8%
(1/128)	
59.4%
(76/128)	35.2%
(45/128)	0.8%
(1/128)	
The extant T. triton	100% (63/63)	98.4% (62/63)**	1.6%
(1/63)	0 (0/63)	
44.4%
(28/63)	49.2% (31/63)	4.8%
(3/63)	
Notes:

* I, unbranched, connected to the lingual tooth edge; II, bifurcated, with one branch connected to the lingual tooth edge and the other to the metaconid; III, trifurcated, with branches connected to the lingual tooth edge, metaconid, and junction of the hypoconid and entoconid, respectively; IV, unbranched, connected to the metaconid; V, having a free end.

** Connected to the lingual tooth edge.

As observed in upper molars, lower molars also demonstrate variations in some structures among a limited number of specimens. For example, the m3 of NWUV 1489.e169 exhibits an ectomesolophid, the sole exception in all lower molars. Moreover, in this particular specimen, the mesolophids of both m1 and m2, along with m3, bifurcate into two branches, representing a unique morphology not observed in any other specimens. Furthermore, some morphotypes, such as the double-branched anterolophulid on m1, the mesolophid of m2 connected to the entoconid, and the III and V morphotypes of mesolophid of m3, can also be viewed as morphological variations due to their exceptional rarity.

Molar measurements are provided in Table 2 and Datasets S5 and S7.

Discussion

Identification of the large-sized hamster material from Syz 2

The taxonomies of Cricetinae fossils from Quaternary deposits in China and extant Chinese Cricetinae species remain highly debated. Based on our observations and recent research advancements (e.g., Lebedev et al., 2018; Wang, Wu & Qiu, 2020), we preliminarily suggest that the inclusion of the following 12 genera in the Chinese Cricetinae, ranging from the beginning of the Quaternary to the present (listed in chronological order of naming; in parentheses are the common junior synonyms): Cricetus Leske, 1779; Cricetulus Milne-Edwards, 1867; Urocricetus Satunin, 1902; Phodopus Miller, 1910; Tscherskia Ognev, 1914 (=Cricetinus Zdansky, 1928, Cansumys Allen, 1928); Allocricetus Schaub, 1930; Sinocricetus Schaub, 1930; Allocricetulus Argyropulo, 1932; Neocricetodon Schaub, 1934 (=Kowalskia Fahlbusch, 1969); Bahomys Chow & Li, 1965; Amblycricetus Zheng, 1993; Nothocricetulus Lebedev et al., 2018. Except that the relationship between Tscherskia and Cricetinus will be discussed in detail below, providing detailed justifications for our conclusions is beyond the scope of this article. Among the mentioned genera, Allocricetus, Sinocricetus, Neocricetodon, Bahomys and Amblycricetus are extinct, while the remaining seven are extant. Among the living genera, Allocricetulus and Nothocricetulus only have very scarce and doubtful fossil records (Cai et al., 2004, 2013), whereas Cricetus and Urocricetus currently have no known fossil records in China.

Aside from Tscherskia, the large-sized hamster material from Syz 2 exhibits distinct differences when compared to other genera listed above. The Syz 2 material can be distinguished from nearly all of these genera by characters such as on m1-2 mesolophids being present but rarely reaching the lingual margin of the teeth, M3 possessing an axioloph, the degree of alternating of the opposite main cusps on M1-3 very small. Furthermore, unlike Neocricetodon and Amblycricetus, which generally have mesoloph(id)s extending to the tooth edge, the mesoloph(id)s of the larger hamster material from Syz 2 scarcely reach the tooth edge. In contrast to Bahomys and Sinocricetus with comparatively higher crowns, the crowns of remains from Syz 2 are low. The sizes of molars, skulls, and mandibles of the large-sized hamsters from Syz 2 are significantly larger than those of Cricetulus (Figs. 2–5, Figs. S1–S3), Phodopus, Urocricetus, Allocricetulus, and Nothocricetulus, but significantly smaller than Cricetus. Some researchers (Zheng et al., 1985, p.117; Cheng et al., 1996, p.40; Jin et al., 2009, p.178) considered that the absence of the mesolophid on m1-2 of Allocricetus is the key character distinguishing it from Cricetinus (i.e., Tscherskia). However, this feature actually pertains to Cricetulus, not Allocricetus, as Allocricetus may not bear the mesolophid on m1 but can develop it on m2 in some specimens (refer to Table 7). On the other hand, some researchers argued that the most crucial character of Cricetinus (i.e., Tscherskia) is the undivided anteroconid of m1 (Kretzoi, 1959; Hír, 1996a, 1997), while that of Allocricetus and Cricetulus is almost always well divided (Hír, 1994, 1996a). However, observations of the extant T. triton molars have shown that the degree of separation of the m1 anteroconid in numerous specimens is comparable to that seen in Allocricetus according to Hír (1994, Fig. 4). In Cricetulus, the separation degree of the anteroconid of m1 in the type species C. barabensis is indeed small, while C. longicaudatus exhibits a well-divided anteroconid of m1 (Fig. S3).

Meanwhile, the great similarity between the large-sized hamster material from Syz 2 and the extant Tscherskia (i.e., T. triton) is readily apparent (Figs. 2–5, Figs. S1–S3). The molar dimensions of the former closely align with those of the extant T. triton, with some measurements even being identical (Table 2). Morphologically, the characters of molars and skulls of the former, such as the degree of alternating of the opposite main cusps on M1-3 small, the anterocone of M1 deeply bifid posteriorly with nearly equal-sized buccal and lingual cones, the mesolophs of M1-3 connected to the metacone instead of being free, M3 with the axioloph, the anteroconid of m1 undivided or weakly divided, the mesolophids of m1-2 present but rarely reaching the lingual margin of teeth, nearly all m3s with well-developed mesolophids, and the interparietal pentagonal, also closely resemble those of the extant T. triton. Therefore, we can safely refer the large-sized hamster remains from Syz 2 to T. triton.

In most skull and mandible measurements, however, the mean values of the material from Syz 2 are lightly larger than those of the extant T. triton (Tables 1 and 2), although the measurements of the single upper and lower molar from both the former and the later are nearly identical (Table 2, Fig. 6). As will be demonstrated below, there are also minor differences in molar morphology between the Syz 2 material and the extant species. Therefore, considering these disparities, it may be more reasonable to further classify these materials from Syz 2 as a chronosubspecies of T. triton, i.e., T. triton varians comb. nov. (=Cricetinus varians, see below for details). In addition, the mean values of the lengths of upper and lower toothrows (M1-3 and m1-3) of the Syz 2 material are also lightly greater than those of the extant T. triton (Table 2). However, the measurements of the single molar imply that this phenomenon, and even certain skull and mandible measurements, may likely result from the burial deformation (see discussion in Xie, Zhang & Li, 2021).

Figure 6 Scatter diagrams of lengths and widths of the first molars of “Cricetinus” varians and Tscherskia triton.

Data source refers to Table 2. The boxes in the figure represent the upper and lower bounds of the length and width values of specimens from specific sites, as the raw data of individual specimen measurements are not available in the original references.

The structure “axioloph” warrants further elaboration here. Both the M3s of T. triton varians from Syz 2 and the extant T. triton possess an anteroposteriorly directed axioloph, which departs from the junction of the protolophule I and the anterior arm of protocone, and forms a groove between itself and the protocone (Figs. 4 and S3). In fact, this structure seems to have been noticed by Zdansky (1928) and Schaub (1930) in the syntypes of T. triton varians from Locality 1 of Zhoukoudian. The term “axioloph,” along with several other terms, was first introduced by Freudenthal & Daams (1988, p.137) to facilitate descriptions of M3 of cricetids. They defined the axioloph as “an axial connection between paracone and hypocone, fundamentally composed of the posterior protolophule and the posterior part of the (ancient) entoloph.” Morphologically, the axiolophs of M3s of Syz 2 specimens and the extant T. triton are obviously distinct from the protolophule IIs of the small-sized hamster remains from Syz 2, and even from all other extant Cricetinae taxa, as their protolophule IIs depart from the posterior wall of the paracone and extend in the anteromedial direction, not forming a groove between itself and the protocone (Fig. S3). By contrast, fossil Cricetinae taxa from Eurasia since the Late Miocene appear to more frequently develop an axioloph on M3, especially in the genus Neocricetodon (=Kowalskia), such as N. moldavicus (see Sinitsa & Delinschi, 2016), N. hanae (see Qiu, 1995), N. yinanensis (see Zheng, 1984b), and N. lii (see Zheng, 1993). This seems to imply a close affinity between Neocricetodon and Tscherskia, although the axioloph is also present in some other genera, such as Nannocricetus primitivus (Zhang, Zheng & Liu, 2008), and seems more often present in cricetid genera of older geologic age (before the late Miocene), such as Democricetodon and Megacricetodon. The phylogenetic significance of the axioloph will not be better understood until a comprehensive phylogenetic analysis covering the taxa mentioned above is conducted, and the homologous structure of the axioloph itself also requires further investigation.

Discussion on the validity of Cricetinus and Cricetinus varians

When Zdansky (1928) established Cricetinus and Cricetinus varians, he solely relied on the skull specimens of extant Cricetus cricetus and Cricetulus phaeus (now considered a subspecies of Nothocricetulus migratorius) for comparison. Consequently, he apparently did not have the opportunity to notice the obvious similarity in molar morphology between the fossils from Locality 1 of Zhoukoudian and the extant T. triton. Zdansky (1928, p.57) seems to have been aware of the potential limitation of his study due to the limited number of extant specimens available for direct comparison with the fossils, so he states in the monograph that “maybe later a generic identity with one of these (extant) genera will result” (translated from German). As expected, doubts about the validity of the genus and species soon emerged. Schaub (1930, 1934) noticed the close resemblance between C. varians and T. triton in molar morphology, but still retained the independent status of C. varians. Teilhard de Chardin (1940, p.56) concluded that he “failed to detect any difference between a ‘Cricetinus’ dentition and the dentition of f.i. Tcherskia in North China”. Teilhard de Chardin & Pei (1941) reiterated that aside from the somewhat larger size, the large-sized hamster fossils from Locality 13 of Zhoukoudian (early Middle Pleistocene in age) showed no appreciable difference from T. triton in either skull or tooth morphology, and they maintained the specific name “varians” for the Pleistocene form primarily due to “geologic convenience.” Zheng & Han (1993) argued that it was challenging to distinguish C. varians from extant T. triton in North China and Northeast China based on size and molar morphologies. Despite these doubts, a large number of such hamster remains discovered in Pleistocene deposits of China were ultimately assigned to C. varians. Meanwhile, as previously mentioned, new fossil hamster species from the Pliocene and Pleistocene deposits of Eurasia have continuously been referred to Cricetinus since Kretzoi (1959). Therefore, it is necessary to clarify the issue of validity of Cricetinus and C. varians.

To address the validity of Cricetinus, the validity of its type species, C. varians, must be considered first. However, the material Zdansky (1928) utilized for establishing C. varians is not only scarce, but also accompanied by a relatively simple description and unclear plates. All of these make it difficult to compare them with T. triton directly. Fortunately, Zheng (1984a) revised most of the hamster fossils collected from the Zhoukoudian area, including C. varians specimens from Locality 1 (type locality) and Localities 3, 9, 13, 15, enabling detailed comparisons with these materials. Except for the material from Zhoukoudian, the specimens from other fossil sites in China that have yielded abundant C. varians fossils were also compared. In the following discussion, we will conduct a detailed comparison of skull and tooth morphologies between C. varians and extant T. triton.

Comparisons of the skull morphologies between C. varians and T. triton

When Zheng (1984a) revisited the hamster fossils from Zhoukoudian, he proposed several distinguishing skull characters to differentiate between C. varians and extant T. triton. However, Xie, Zhang & Li (2021) assessed these characters proposed by Zheng (1984a) and concluded that these differences between C. varians and T. triton skulls were questionable and required further verification. Therefore, it is not necessary to reiterate them here.

Topachevsky & Skorik (1992, p.181) suggested three morphological differences in skull features between Cricetinus and Tscherskia. Based on the context, these opinions appear to be founded only on the observation of the holotype (a maxillary fragment with M1-3) of Cricetinus gritzai, rather than the specimens of the type species (C. varians) of the genus. Firstly, Cricetinus is said to differ from Tscherskia by having a wider and more concave masseteric plate (i.e., “zygomatic plate” in the present article). However, Topachevsky & Skorik (1992) did not provide any measurements of the zygomatic plates of Cricetinus and Tscherskia to substantiate this claim, even though the degree of depression of the surface of the zygomatic plate seems challenging to quantify. Even if this assertion holds, a wider and more concave zygomatic plate may only be a feature of the Cricetinus gritzai species, not the entire Cricetinus genus, because our observations show no obvious difference in the characters of the zygomatic plate between Tscherskia triton varians from Syz 2 and living T. triton (Figs. 2 and S1). Secondly, Cricetinus is said to develop stronger ridges along the posterior side of the incisive foramina (the rim of the area for the lateral masseter?) than Tscherskia. However, we likewise failed to discern any appreciable difference in the ridges between T. t. varians from Syz 2 and extant T. triton (Figs. 2 and S1). Thirdly, the position of the masseteric tuberosity in Cricetinus is considered more similar to that in Cricetus than in Tscherskia. Our observations show that the position of the posterior margin of the masseteric tuberosity in living Cricetus cricetus (closer to the posterior edge of the incisive foramen) seems to be slightly further back than that in living Tscherskia triton (closer to the middle of the incisive foramen). The position of the masseteric tuberosity of T. t. varians from Syz 2 more closely resembles that of extant T. triton rather than C. cricetus (Figs. 2 and S1). In conclusion, since the three distinguishing characters between Cricetinus and Tscherskia proposed by Topachevsky & Skorik (1992) seem to be based on just one specimen of C. gritzai (the holotype), and we were unable to detect the aforementioned differences between extant T. triton and T. t. varians from Syz 2, the validity of these differences, in our opinion, is questionable.

Comparisons of the teeth morphologies between C. varians and T. triton

Comparisons of the teeth size

Table 2 and Fig. 6 show the measurements and scatter diagrams of C. varians from Zhoukoudian in Beijing (Zheng, 1984a), Jinniushan in Liaoning Province (Zheng & Han, 1993), and Renzidong in Anhui Province (Jin et al., 2009), as well as T. triton from Syz 2 and extant T. triton. It is evident that, with the exception of the material from Renzidong that is significantly smaller, the average molar sizes from other localities are quite similar, and the data ranges also substantially overlap. In other words, we cannot differentiate C. varians from T. triton based on their size. As to the material from Renzidong, its significantly smaller size and markedly older geologic age—approximately 2 Ma (Jin, Qiu & Zheng, 2009) compared to the Middle Pleistocene age of other localities—cast doubt on its identification as C. varians. It is possible that the material from Renzidong represents a new form.

Additionally, with the reassignment of hamster material initially identified as Cricetinus varians (or Cricetinus cf. varians, Cricetulus (Cricetinus) varians) from several Early Pleistocene sites in China, such as Localities 12, 18 of Zhoukoudian in Beijing, and Gongwangling in Lantian, Shaanxi, being assigned to the genus Allocricetus (Zheng, 1984a), East cave of Zhoukoudian has become the only Early Pleistocene site in China, besides Renzidong, yielding Cricetinus varians fossils. However, the length of M1-3 of Cricetinus varians from East cave is merely 4.83 mm (Cheng et al., 1996, Tables 3–11, p.40), smaller than the lower limit of the variation range for that measurement in “typical” C. varians and extant Tscherskia triton (Table 2). More importantly, the m1s of C. varians from East cave completely lack the mesolophid (Cheng et al., 1996, p.40), which markedly differs from “typical” C. varians and extant T. triton (Table 3). Therefore, the material identified as C. varians from East cave necessitates reevaluation of its classification. Given the above explanations, except Tscherskia sp. from the Late Pliocene Youhe fauna (Xie, Zhang & Li, 2021), there is now no reliable fossil of Tscherskia in China predating the Middle Pleistocene.

Comparisons of the teeth structures

In a hamster individual, the molars symmetrically distributed in the oral cavity (e.g., the left and right M3) may exhibit minor morphological differences; therefore, the morphological structures of both the left and right molars of large-sized hamsters from Syz 2 and living T. triton were statistically analyzed in the present study. The material of C. varians used for comparison here is mainly from Zhoukoudian (Zheng, 1984a) and Jinniushan (Zheng & Han, 1993).

m1: In extant T. triton, about 30% of specimens have a mesolophid (Table 3). The mesolophids are consistently weakly developed and of very short length, with the longest mesolophid not exceeding 1/5 of the distance from the base to the edge of the tooth. In most cases, the mesolophid only appears as a small bulge. It is either connected to the metaconid (9.5%, 2/21) or has a free end (90.5%, 19/21).

The localities in Table 3 are listed in descending order, approximately following their geological age from oldest to youngest (ZKD Loc.1, ca. 0.6–0.2 Ma; Jinniushan, ca. 0.31–0.2 Ma; ZKD Loc.3, late Middle Pleistocene; Syz 2, ca. 0.2 Ma). Although the frequencies of mesolophids in C. varians and T. triton differ across various geological ages, there is no evident discontinuity between them, and as the age advances, the frequency of the mesolophid decreases. Given the similarities in other aspects of tooth morphology and the practicality of classification, it is more appropriate to interpret the differences in mesolophid frequencies as a result of gradualistic evolution within a single species, specifically the progressive reduction of the mesolophid, rather than interspecific or intergeneric differences.

m2: Table 4 presents the frequencies of mesolophids on m2s of T. triton and C. varians. As shown in the table, throughout their geologic history, the frequencies of mesolophids on m2s in both T. triton and C. varians were consistently high and similar, although slightly lower in extant T. triton. A comparable pattern is also observed in the proportions of morphotype III in C. varians (excluding Jinniushan specimens) and extant T. triton. Therefore, the mesolophid morphology on m2s in T. triton and C. varians further substantiates the congruence of the “two” species, and it appears more plausible to interpret the minor differences between the two as a gradualistic evolution within a single species, specifically the reduction of the mesolophid, similar to the situation observed in mesolophids on m1.

m3: Table 5 shows the frequencies of mesolophids on m3s of T. triton and C. varians. As seen from the table, mesolophids are present in nearly all specimens. The proportions of the mesolophids morphotype “connected to the lingual tooth edge” are consistently high, but no clear regularity emerges. Comparing the proportions of more detailed morphological structures is difficult due to insufficient data. But overall, the characters of m3s of T. triton and C. varians are still quite consistent.

M1: The lingual anterocones and protocones on M1s of T. triton and C. varians are consistently connected by an anterolophule, whereas the anterolophule behind the buccal anterocone is not always present. Table 6 illustrates that the occurrence frequency of the anterolophule behind the buccal anterocone is high in both T. triton and C. varians. However, as the statistical data for C. varians are based on a relatively small number of specimens, the reliability of the comparison is diminished. The frequencies of “protolophule I” are unstable and appear to lack any discernable regularity.

Table 6 Comparisons of anterolophules and protolophules I of M1s between Tscherskia triton and “Cricetinus” varians.

Species and localities	Frequencies of anterolophules behind the buccal anterocone	Frequencies of protolophule Is	
C. varians of ZKD Loc. 1, 3	—	76%	
C. varians of JNS	100% (9/9)	≥30%	
T. triton varians of Syz 2	89.1% (41/46)	57.4% (27/47)	
The extant T. triton	71.6% (53/74)	37.2% (32/86)	

M2 and M3: There is practically no morphological difference between M2s and M3s of T. triton and C. varians.

In summary, C. varians and T. triton exhibit substantial consistency in skull and tooth morphologies. Although minor differences in tooth morphology exist between them, these differences exhibit continuous variation and can only be discerned with statistically abundant material. Therefore, we propose that C. varians should be considered a chronosubspecies of T. triton, i.e., T. triton varians comb. nov., and Cricetinus should be regarded as a junior synonym of Tscherskia.

Referred species of Tscherskia

Apart from Cricetinus varians, there are six other species in Eurasia that have been referred to Cricetinus:

Cricetinus europaeus Kretzoi, 1959. The type locality of this species is Csarnóta 2 in Hungary. The majority of researchers believe that the geological age of this site is MN 15 (Venczel & Gardner, 2005). The type specimens of C. europaeus consist of only three molars, but one M2 among these three molars was later identified as C. janossyi by Hír (1996b). Hír (1994) discovered additional materials for this species and described them in detail when examining the materials from the type locality, thus clarifying the nature of the species. Although C. europaeus is among the earliest Cricetinus species in Europe (Hír, 1994), it appears to exhibit rather advanced morphologies. For instance, the presence ratios of mesolophids on m1 and m2 are even lower than those of extant T. triton (Table 7); however, due to the scarcity of material, this observation requires further validation with additional material in the future.

Cricetinus gritzai Topachevsky & Skorik, 1992. The type locality of this species is Odessa Catacombs, Ukraine, with an age of MN15. A notable character of this species is that all m1s and partial m2s possess a mesolophid (Koufos et al., 2001). On one hand, this feature indicates a more primitive nature (in other Cricetinus or Tscherskia species, the mesolophid frequency of m1 reaches a maximum of 70%). On the other hand, the character itself is also unique, because in cricetids, the mesolophid frequency of m1 is almost always lower than that of m2, whereas in this species the situation is reversed. Moreover, other molars of C. gritzai are slightly smaller in size than those of T. triton, but only M3 is considerably larger than that of T. triton (Topachevsky & Skorik, 1992). If this discrepancy is not a statistical error (given that there is only one M3), it may also illustrate the primitive nature of C. gritzai.

Cricetinus beremendensis Hír, 1994. The type locality of this species is Beremend 15 in Hungary, with a geologic age of 2.7 Ma (Hír, 1994; Pazonyi, 2011). The molar morphology of this species, particularly the degree of mesolophid development, is markedly different from other species currently classified in Cricetinus, but closely resembles Allocricetus ehiki and A. bursae in size and morphology (Table 7). Thus, it seems more reasonable to assign this species to Allocricetus Schaub, 1930.

Cricetinus janossyi Hír, 1996. The type locality of this species is Osztramos 7 in Hungary, with a geologic age of approximately 2.3 Ma (Hír, 1996b; Pazonyi, 2011). The molar morphology of this species is very similar to that of T. triton varians from Syz 2 (Table 7), although the former is slightly larger in size, and their ages are significantly different. C. janossyi is also among the earliest Cricetinus species in Europe, first appearing in Csarnóta 2 of Hungary at the same time as C. europaeus.

Cricetinus koufosi Koliadimou, 1996. The type locality of this species is Ravin Voulgarakis in Mygdonia basin of Greece (Koufos et al., 2001). The age of Ravin Voulgarakis has been dated to the Nagyharsanyhegy phase of the Biharian (ca. 1.2–0.7 Ma) (Koufos et al., 2001), making this species the latest among several Cricetinus species in Europe. Additionally, this species has been discovered in Marathoussa of Mygdonia basin, with the age of the locality dated to the Betfia phase of the Biharian (ca. 1.5–1.2 Ma) (Koufos et al., 2001). Many molar characters of this species remain unclear, but the absence of the mesolophid on m1 may indicate its relatively advanced nature.

Cricetinus mesolophidos Wu & Flynn, 2017. Xie, Zhang & Li (2021) concluded that it was more reasonable to place C. mesolophidos in Neocricetodon rather than in Cricetinus (i.e., Tscherskia).

In summary, we suggest that C. europaeus, C. gritzai, C. janossyi, and C. koufosi should be transferred to Tscherskia, while C. beremendensis should be transferred to Allocricetus, and C. mesolophidos to Neocricetodon. However, this treatment is provisional, because the characters of some of these species remain unclear. Except the type species T. triton, the type localities of the other four Tscherskia species are situated within a relatively small area covered by several neighboring countries in southeastern Europe. This significant geographic distance between T. triton and other species introduces a degree of uncertainty to the above classification (Kretzoi, 1959; Hír, 1994). In addition, Storch (1974) described a species T. rusa from the Holocene (dated between 2200–700 B.C.) of northern Iran, whose geographic location and age are highly perplexing. Although we have tentatively placed it in Tscherskia, the validity of this species and whether it should be referred to Tscherskia clearly warrant further investigation. Table 7 provides a summary of comparisons of frequencies of mesolophids on lower molars among the species of Cricetinus, Tscherskia, and some related genera (Cricetulus, Nothocricetulus, and Allocricetus).

Origin and dispersal of Tscherskia

Zheng (1984a, 1984b), Zheng et al. (1985), and Zheng & Han (1993) suggested that Cricetinus (a junior synonym of Tscherskia) very likely originated from the genus Kowalskia (a junior synonym of Neocricetodon), an idea tentatively proposed by Fahlbusch (1969). Qiu & Li (2016) remarked that this viewpoint merits further investigation. We also concur with this viewpoint, and the reasons for this deduction have already been fully explained by Zheng (1984b) (as discussed above, the presence of the axioloph on M3 in both genera also appears to support this), so it is not necessary to reiterate these points here.

The question that arises now is: when and where (Asia or Europe) did Tscherskia originate? Based on current evidence, the earliest appearance of Tscherskia in Europe predates that in Asia. The earliest species of Tscherskia in Europe, T. europaeus and T. janossyi, both emerged at Csarnóta 2 (MN 15, ca. 5–3.5 Ma) in Hungary (Hír, 1994; Venczel & Gardner, 2005). In Asia, the earliest known Tscherskia is T. sp., represented by a fragmentary mandible with m2-m3 that is discovered from Youhe Formation (ca. 3.40–2.59 Ma, Yue & Xue, 1996) in Linwei District, Shaanxi Province, China and belongs to the Youhe fauna (Xie, Zhang & Li, 2021). However, as previously noted, all other credible materials of Tscherskia found in China are from the Middle Pleistocene or later. This nearly empty fossil record of Tscherskia in China or East Asia before the Middle Pleistocene is a major challenge to the idea of an East Asian origin for Tscherskia, although species morphologically similar to Tscherskia triton have been found in the Late Pliocene (?) in China (e.g., Neocricetodon yinanensis). Furthermore, Europe exhibits a higher diversity of Tscherskia species compared to Asia.

Therefore, based on the available evidence, it seems more probable that Tscherskia originated from Neocricetodon during the Early Pliocene in Europe and subsequently spread to Asia. It is possible that another dispersal event in the same direction occurred during the late Early Pleistocene, which could account for the absence of credible Tscherskia fossils in China throughout the Early Pleistocene. Meanwhile, the Tscherskia that arrived in East Asia during the first dispersal event likely became extinct shortly thereafter and did not survive into the Pleistocene. Of course, this hypothesis still requires verification through the examination of additional material in the future.

Conclusions

The detailed morphological description and comparative study of hundreds of large-sized hamster remains collected from the late Middle Pleistocene Locality 2 of Shanyangzhai (Syz 2) indicate that they should be referred to a chronosubspecies of the extant Tscherskia triton—T. triton varians comb. nov. This chronosubspecies is highly similar to extant T. triton in size and most molar characters, but exhibits slightly higher frequencies of mesolophids on m1 and m2. In most skull and mandible measurements, the mean values of the former may be lightly greater than those of the later. To resolve the longstanding debate over the validity of Cricetinus Zdansky, 1928 and C. varians Zdansky, 1928, a detailed comparison of skull and molar morphology was conducted between C. varians and T. triton. The results demonstrated that the differences between the two are very slight; thus, C. varians can only be treated as a chronosubspecies of T. triton, i.e., T. triton varians comb. nov., and Cricetinus should be considered a junior synonym of Tscherskia. We tentatively propose that among the seven species once referred to Cricetinus in Eurasia, C. europaeus, C. gritzai, C. janossyi, and C. koufosi should be transferred to Tscherskia, while C. beremendensis should be transferred to Allocricetus, and C. mesolophidos to Neocricetodon. Apart from Tscherskia sp. from the Late Pliocene Youhe fauna, there are no credible fossils of Tscherskia in China earlier than the Middle Pleistocene. Based on the current evidence, Tscherskia may have originated from Neocricetodon during the Early Pliocene in Europe and subsequently spread to Asia, with T. triton being its sole extant representative now only inhabiting East Asia.

Supplemental Information

Supplemental Information 1 The skulls of extant Tscherskia triton, Cricetulus barabensis and Cricetulus longicaudatus.

(A) Tscherskia triton, catalogue number: 礼122; (B) Tscherskia triton, catalogue number: 齐070; (C) Cricetulus barabensis, catalogue number: 礼151; (D) Cricetulus longicaudatus, catalogue number: 礼011. (A1), (B1), (C1), (D1), dorsal view; (A2), (B2), (C2), (D2), ventral view; (A3), (B3), (C3), (D3), lateral view; (B4), anterior view. The underlined label indicates the image has been reversed.

Click here for additional data file.

Supplemental Information 2 The mandibles of extant Tscherskia triton, Cricetulus barabensis and Cricetulus longicaudatus.

(A) Tscherskia triton, left mandible, catalogue number: 礼094; (B) Tscherskia triton, right mandible, catalogue number: 齐070; (C) Cricetulus barabensis, left mandible, catalogue number: 148b; (D) Cricetulus longicaudatus, left mandible, catalogue number: 瓦002. (A1), (B1), (C1), (D1), lingual view; (A2), (B2), (C2), (D2), buccal view; (A3), (B3), (C3), (D3), occlusal view. (A) and (B) share one scale bar; (C) and (D) share one scale bar.

Click here for additional data file.

Supplemental Information 3 The molars of extant Tscherskia triton, Cricetulus barabensis and Cricetulus longicaudatus.

(A), (B), (C), (D), Tscherskia triton; (E), (F), Cricetulus barabensis; (G), (H), Cricetulus longicaudatus. (A) right upper molars, catalogue number: 158; (B) right upper molars, catalogue number: 127; (C) left lower molars, catalogue number: 齐060; (D) left lower molars, catalogue number: 齐070; (E) left upper molars, catalogue number: 礼144; (F) left lower molars, catalogue number: 礼144; (G) left upper molars, catalogue number: 027; (H) left lower molars, catalogue number: 027. The underlined label indicates the image has been reversed.

Click here for additional data file.

Supplemental Information 4 Raw data of measurements of specimens examined.

All raw data is in mm.

Click here for additional data file.

We would like to express our sincere appreciation to Prof. Qiu Zhuding for his assistance in improving the manuscript, to Prof. Zheng Shaohua for valuable discussions, and to Prof. Wu Wenyu for her warm support (all affiliated with the Institute of Vertebrate Paleontology and Paleoanthropology, Chinese Academy of Sciences, China). We are grateful to the editor, Dr. Kenneth De Baets, and reviewers Drs. Maxim Sinitsa, János Hír, and Jordi Agustí for their insightful comments, which substantially enhanced the original manuscript. We also extend our thanks to Li Zhixuan (College of Life Sciences, NWU), Wang Kaifeng and Wang Yan (Shaanxi Institute of Zoology, China), Zhang Lixun and Liao Jicheng (School of Life Sciences, Lanzhou University, China), and Zhang Yanming (Northwest Institute of Plateau Biology, Chinese Academy of Sciences, China) for their friendly assistance during the examination of extant hamster specimens. Additionally, we thank Prof. Robert F Diffendal, Jr. for his linguistic help.

Additional Information and Declarations

Competing Interests

Author Contributions

Data Availability

The authors declare that they have no competing interests.

Kun Xie conceived and designed the experiments, performed the experiments, analyzed the data, prepared figures and/or tables, and approved the final draft.

Yunxiang Zhang conceived and designed the experiments, authored or reviewed drafts of the article, and approved the final draft.

Yongxiang Li conceived and designed the experiments, authored or reviewed drafts of the article, and approved the final draft.

The following information was supplied regarding data availability:

The raw measurements and specimen numbers are available in the Supplemental Files.

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
