# Peer review of "Large-sized fossil hamsters from the late Middle Pleistocene Locality 2 of Shanyangzhai, China, and discussion on the validity of Cricetinus and C. varians (Rodentia: Cricetidae)"

_PeerJ, doi:10.7717/peerj.15604_

## Round 0.1 · original submission · Minor Revisions

You provide a timely revision of taxa previously assigned to Cricetinus and Tscherskia as well as supporting description of material you assign to the subspecies of the Greater long-tailed hamster from the Pleistocene of Hebei Province. All agree unanimously agree that it is a valuable contribution to the field and is of great interest to be published. However, some crucial points need be clarified before publication. The main points being:

Comparative analysis: I agree with reviewer 1 that the paper would benefit from direct comparisons between extant and fossil forms at least illustrating cranial/dental/mandibular specimens of living Tscherskia triton as well as ideally other closely related extant and extinct species of closely related cricetines. Also, a more-in depth discussion of the cranial morphology of Tscherskia and Cricetinus is needed (compare reviewer 1). Tscherskia rusa Storch should be included in the referred species to this genus (line 141-143; compare reviewer 3). You are quite critical of Hír (lines 158-159), but it isn’t clear if you studied the Hungarian material directly. Please rephrase and clarify.

Age and interpretations of the Hungarian material: Please update the age assignment of the Csarnóta 2 and Osztramos 7 fauna to 3,5 Ma and 2,5 Ma, respectively (compare reviewer 3). Please also cite more recent supporting evidence for these age assignments. You also state that Tscherskia was relatively common during the Pliocene in southeastern Europe in line 648 but this should be rephrased as Cricetinus or Tscherskia are relative rare elements in Csarnóta and Osztramos 7 (compare reviewer 3).

Origination and relationships of Tscherskia: you present the hypothesis that Tscherskia likely originated from Kowalskia in Europe and only subsequently spread to Asia. In this context, it is crucial to also the discuss the potential inconsistency with climatic development and the shortening of mesolophs/mesolophids (compare reviewer 3).

Phylogenetic analysis: Your revision of Tscherskia relies on craniodental morphology and assumptions on the phylogenetic meaning of these characters which would benefit from a supporting morphological phylogenetic analysis. I agree with reviewer 1 that such an analysis would make your interpretations more robust.

Typographical errors and formatting inconsistencies: reviewers 1, 2 and 3 reported various typographical errors and inconsistencies which all need to be clarified before publication (see also annotations in pdfs). As pointed out by reviewer 2, the genus Kowalskia (a junior synonym of Neocricetodon) is variously mentioned in the text, either as Kowalskia, Kowalskia (= Neocricetodon) or Neocricetodon (= Kowalskia). This should be homogenized. I agree with reviewer 3 that the use of “terminates” (line 288), “degraded” (line 317) and “degenerate” (lines 555, 558) needs to be revised as well as Archyropulo versus Argyropulo (line 405 and Csartóna versus Csarnóta (lines 585, 647).

References: some references are missing from reference list (e.g., Koliadimou 1996, Kishida 1929) or inconsistently cited in text (Jin et al. 2009 a versus b; compare reviewer 3). Please make sure all references are consistently addressed in text and references list.

Figures: Please provide enlarged, higher resolution and clearer figures of cranial/mandibular/skull specimens in Figures 2 and 3 reducing vast, uninformative areas as much as possible. The clarity would benefit from labelling the dental and cranial structures discussed in the text (compare reviewer 1). As pointed out by Reviewer 2, you largely follow Freudenthal et al. nomenclature for description but not entirely. Some terms are confusing and including the terms in figures would clarify the used terminology.

Please address these and all other points raised in the reviews including annotated pdfs. I look forward to receiving the revised manuscript.

·

Basic reporting

Basic Reporting
Kun Xie, Yunxiang Zhang, and Yongxiang Li describe the fossil material belonging to the extinct subspecies of the living Greater long-tailed hamster (Tscherskia triton varians) from the mid-Pleistocene cave deposits of Shanyangzhai (Syz 2) in Hebei Province, China. The core of the article, however, is a taxonomical revision of hamster taxa previously allocated to the extinct genus Cricetinus. These medium- to large-sized hamsters, initially described from the worldwide-known Locality 1 of Zhoukoudian, are among the most widespread Pliocene Eurasian cricetines with a large geographical area and unclear phylogenetic relationships with other hamsters. The strength of the article is an attempt to clarify this point. The main conclusion of the study is rather straightforward: Cricetinus represents the junior synonym of Tscherskia, and most species traditionally allocated within Cricetinus are transferred to Tscherskia, whereas the type species of the genus (Cricetinus varians) are regarded as the subspecies of Tscherskia triton. This is a long-awaited study with the data and conclusions seem to be sound. The manuscript is relatively well-written, mostly clear and concise, although multiple misprints and inconsistencies are present. All of them can and should be corrected (I made comments to them in the attached file).
The figures are mostly clear and concise. This is especially the case of the teeth specimens, which, albeit not coated to prevent obscuring of the occlusal surface structures by pigmented spots, look nice and essentially clean. the graphs are also good. Unfortunately, the cranial and mandibular specimens are not as good. I think the excellent skull specimens are shown (scaled) too small and surrounded by a vast blank (white) space. This certainly hampers the appreciation of the specimens and their morphology. Thus, I recommend the authors alter figures 2 and 3 by reducing the vast, uninformative areas as much as possible. Also, it would be good to label the morphological structures (both dental and cranial) discussed in the text. My main complaint, however, is a total lack of a comparative approach in the figures. Considering that the study centers around the question of the taxonomic validity of Cricetinus and its relationships with other close-related taxa, I strongly recommend adding figures illustrating cranial, dental, and mandibular specimens of living Tscherskia triton in comparison with "Cricetinus" varians. Ideally, this should be a comprehensive figure comparing species of Cricetinus with an array of extant and extinct close-related cricetines.

Experimental design

The research contains no experimental materials. However, the data presented in the manuscript is original. The interpretations and conclusions are straightforward. In general, the research fits with the scope of the study.

Validity of the findings

I have already commented on the strengths and some weak points of the article. The discussion was for the most part easy to follow. However, another potential weakness of the research is related to this section. The discussion covers several aspects of craniodental morphology of Tscherskia, and makes taxonomic assumptions regarding the phylogenetic meaning of these characters. Such assumptions, however, could be tested and substantiated by the result of a phylogenetic analysis. The manuscript, instead, does not present any phylogenetic or numerical analyses. In my personal opinion, such an analysis could be a valuable support to the ideas presented in the study. On another hand, this is just a recommendation that I leave to the Editor's discretion.
Also, I suggest a more in-depth discussion of the aspects of the cranial morphology of Tscherskia and Cricetinus, considering and discussing some arguments and conclusions gathered by the previous authors. In particular, Topachevsky and Skorik (1992, p. 181) pointed that: " Cricetinus clearly differs from Tscherskia in having a wider and more concave masseteric plate of the skull, stronger ridges along the posterior side of the incisive foramina, and position of the masseteric tuberosities more similar to those in Cricetus, than in Tscherskia)"

Additional comments

Overall, I see this manuscript as potentially making a valuable contribution to our understanding of the morphology and taxonomy of extinct greater long-tailed hamsters. After some editing and greater care of comparative and phylogenetic results, I believe the manuscript will be worthy of publication.

·

Basic reporting

Every thing is corret

Experimental design

Everything is correct

Validity of the findings

Everything is correct

Additional comments

This is a very comprehensive and interesting paper which shed considerable light in a significant part of the Cricetid faunas from the Plio-Pleistocene of Asia and Eastern Europe. The paper reviews in an exhaustive way the state of the art relative to the genera Tscherskia and Cricetinus, clarifing in a considerable way the relation between these two taxa. In this way, it merits to be published. However, some corrections are needed before publication:

- The genus Kowalkia (a junior synonim of Neocricetodon) is variuosly mentioned in the text, either as Kowalskia, Kowalskia (=Neocricetodon) or Neocricetodon (=Kowalskia). Please, unify. In my opinion the correct form is Neocricetodon (=Kowalskia).

- 141-143: Include Tscherskia rusa Storch in the referred species to this genus.

- 288: Incisors are evergrowing teeth lacking roots. So it is no sense to tolk about "its roots terminates..." May be, its posterior end... Same comment to 344.

- 317: The use of the term "degraded" to refer to the posterior portion of the M 3 is unappropiate. Use another term (reduced?). The same for 555: "the mesolophid gradually degenerates" or 558: "degenreative trend". Mesolophid cannot "degenerate", may be reduce...

regarding the nomclature used in the descriptions, the authors follow in general Freudenthal et al., but not completely. Some terms like "Protolophule I and II are confusing. Would be fine if the authors include a figure with M 1 and m1, with the terminology used for each cusp and ridge.

In any case, this is a very interesting paper which merits to be publushed.

·

Basic reporting

1. Comprehensive evaluation
Overall the manuscript is a valuable work and I support the publication.

2. Topic selection
It is justified, because the status and validity of the Cricetinus genus is rather enigmatic. The authors give essential new results.

Experimental design

4. Methodology
The applied metrical and statistic morphological methods are referable to the general practice of the research activity in the microvertebrate palaeontology.

5. Artwork and illustrations
The visualization is fundamental in our science. Overall I am satisfied with the quality of the presented figures and tables.

6. Language and style
These are appropriate for the general requirements of a scientific text.
Minor remark:
585, 647: Csartóna Csarnóta

7. Handling of the literature
It is perfect, embraces the complete concerning literature published in Asia and in Europe.
I have only minor remarks.
a. I haven’t found the citation KOLIADIMOU, 1996 in the reference list.
b. Jin et al. 2009 a and 2009 b are not always exact in the text. E.g. 425
c. 405: ARCHYROPULO ARGYROPULO
d. It is clear that the authors published earlier than 1900 does not need to be indicated in the reference list, but Kishida 1929 e.g. can be indicated.

Validity of the findings

3. The studied material
In the microvertebrate palaeontology the statistic amount of the finds is crucial. In this case this requirement is fulfilled.

The conclusions are well founded and I agree with them in overall

Additional comments

8. Remarks for the content


158-159: „ These characters proposed by Hír are not comprehensive, and some of them even differ greatly from the facts.”

Excuse me, it is a bit categoric verdict. Have You studied the Hungarian material directly?

647: „…an age of approximately 4 Ma (Hír, 1994)”
The recent estimation for the numeric age of the Csarnóta 2 fauna is about 3,5 Ma. Osztramos 7 is about 2,5 Ma (Szentesi et al. 2015: Fragmenta Palaeontologica Hungarica, 32: 49-66) (Pazonyi 2011: Acta zoologica cracoviensia 54 A(1-2): 1-29.

654: „Tscherskia seems more likely to have originated from Kowalskia during the early pliocene is Europe and then spread to Asia.”
This idea is interresting, but please take into consideration that Csarnóta 2 and Osztramos 7 faunas are mirrored mild humid climate with glirids, murids, petauristids and the latest eomyids. The european Cricetinus or Tscherskia species adopted for this mild climate. I have doubts that this animal was able to cross the arid and continental Central Asia. So the evolutionary connection of the European and Chinese species is an open question. Maybe a direct comparison of the finds would be a good idea.
The connection of Kowalskia and Cricetinus and Tscherskia is another question, because I don’t know any tendency for the shortening of the mesolophs/mesolophids in the European pliocene Kowalskia materials.

648: „Tscherskia was relatively common during the Pliocene in southeastern Europe”
Please be careful, because the Cricetinus or Tscherskia finds are definitely rare elements of the faunas Csarnóta 2 and Osztramos 7.

I should like to emphasize that my remarks are not question the fundamental values of the manuscript. Those are only points out fine details. This work is valuable and I wholeheartedly support the publication.

26. 01. 2023, Pásztó, Hungary
Prof. János Hír

---

## Round 0.2 · Minor Revisions

Thank for revising your manuscript so swiftly which makes the paper easier to follow and of broader relevance. I would love to see it published, but some minor but crucial changes still need to be made before publication. The main points being:

Combination nova: you are creating a new subspecies and a new genus / (sub)species combination so that adding comb. nov. (or n. comb.) needs to be added in abstract and when changing nomenclature, a Zoobank entry should to be made.

Synonymy list: please provide information on which assignments you could re-investigate directly (v) and which not and remain uncertain (?). See Matthews (1973) for recommendations.

Defining terms: please make sure the terms M1-3 and m1-3 are defined the first time they are used. The meaning might be trivial to some readers but not to all.

Formatting/language/typographical issues: there remains various issues related to formatting or language which need to be resolved (see annotated pdf). I did my best to point out all remaining issues but there might be issues I might have missed as I am not a native speaker or expert on mammal microremain terms. I would therefore advise for a colleague fluent in English to thoroughly proofread the paper one more time before resubmission. Your paper should be able to be understood by all (paleo)biologists.

Please address these and all other points raised in the annotated pdf.

I look forward to receiving the revised manuscript.

Suggested reference:

Matthews, S. C. (1973). Notes on open nomenclature and on synonymy lists. Palaeontology, 16(4), 713-719.

---

## Round 0.3 · accepted · Accept

Thank you for addressing these final suggestions. I also thank you for your patience. I agree with reviewer 2 that the paper is now better structured and easier to follow. There are some final minor suggestions by reviewer 1 that i would like you to implement during the proofing phase (see annotated pdf). I look forward to seeing this paper published.

·

Basic reporting

I find the resubmitted version of the paper better structured, written, and more informative and readable. Now it has much fewer errors and omissions and needs only minor revisions,
Please find attached the file peerj-reviewing-80883-v2-MS_commente.pdf
In my view, the paper deserves publication in PeerJ after being corrected as recommended in the attached PDF. Finally, I congratulate the authors with this interesting and long-awaited research.
Sincerely yours,
Maxim V. Sinitsa

Experimental design

-

Validity of the findings

-

Additional comments

-